# Long-Lasting Epigenetic Changes in the Dopamine Transporter in Adult Animals Exposed to Amphetamine during Embryogenesis: Investigating Behavioral Effects

**DOI:** 10.3390/ijms241713092

**Published:** 2023-08-23

**Authors:** Tao Ke, Ganesh Ambigapathy, Thanh Ton, Archana Dhasarathy, Lucia Carvelli

**Affiliations:** 1Harriet L. Wilkes Honors College, Florida Atlantic University, Jupiter, FL 33458, USA; tke@fau.edu (T.K.); tton2020@fau.edu (T.T.); 2Department of Biomedical Sciences, School of Medicine and Health Sciences, University of North Dakota, Grand Forks, ND 58202, USAarchana.dhasarathy@und.edu (A.D.); 3Stiles-Nicholson Brain Institute, Florida Atlantic University, Jupiter, FL 33458, USA; 4Department of Biomedical Science, Florida Atlantic University, Jupiter, FL 33458, USA

**Keywords:** dopamine transporter, amphetamine, histone methylation, epigenetic modifications, *C. elegans*

## Abstract

The dopamine transporter (DAT) is an integral member of the dopaminergic system and is responsible for the release and reuptake of dopamine from the synaptic space into the dopaminergic neurons. DAT is also the major target of amphetamine (Amph). The effects of Amph on DAT have been intensively studied; however, the mechanisms underlying the long-term effects caused by embryonal exposure to addictive doses of Amph remain largely unexplored. As in mammals, in the nematode *C. elegans* Amph causes changes in locomotion which are largely mediated by the *C. elegans* DAT homologue, DAT-1. Here, we show that chronic embryonic exposures to Amph alter the expression of DAT-1 in adult *C. elegans* via long-lasting epigenetic modifications. These changes are correlated with an enhanced behavioral response to Amph in adult animals. Importantly, pharmacological and genetic intervention directed at preventing the Amph-induced epigenetic modifications occurring during embryogenesis inhibited the long-lasting behavioral effects observed in adult animals. Because many components of the dopaminergic system, as well as epigenetic mechanisms, are highly conserved between *C. elegans* and mammals, these results could be critical for our understanding of how drugs of abuse initiate predisposition to addiction.

## 1. Introduction

The dopamine (DA) transporter (DAT) is a transmembrane protein expressed in the dopaminergic neurons which inactivates DA signaling by binding to DA and reaccumulating DA released in the synaptic cleft back into the dopaminergic neurons. For this reason, DAT is a key factor in modulating the intensity and duration of DA signaling and DA homeostasis. Gene mutations leading to functional impairment or increased expression of DAT have been correlated with various pathological conditions, such as parkinsonism or attention deficit hyperactive disorder (ADHD) [1,2]. Moreover, higher levels of DAT expression are associated with higher oxidative stress via DA oxidation [3]. Importantly, DAT is also the main molecular target of psychostimulants including amphetamine (Amph), methamphetamine and cocaine [4]. Because of the chemical structure similarities between Amph and DA, DAT binds to Amph and quickly moves Amph into the dopaminergic neurons. By doing so, Amph prevents DA from being cleared from the synaptic cleft and, thus, it increases the concentration of extracellular DA. Once inside the neurons, Amph depletes the dopaminergic vesicles by interacting with the vesicular monoamine transporter, and the resulting increased levels of intracellular DA promote DA efflux via reversal of DAT, i.e., DA is moved from the neurons into the synaptic cleft.

Since its discovery, Amph has been used therapeutically to treat a variety of disorders, including narcolepsy and attention deficit disorders. However, the highly addictive properties of Amph limit its use. The psychostimulant and addictive properties of Amph are primarily caused by its ability to increase synaptic DA in the brain via both augmenting vesicle fusion [5] and/or DAT reversal transport and by preventing DAT-mediated DA uptake [4]. Several studies have extensively investigated the role of DAT in mediating Amph-induced behaviors and have highlighted the critical role played by DAT in the rewarding and locomotor effects of Amph [6,7]. For example, animals lacking expression of DAT exhibit hyperactivity, cognitive deficits and sleep dysregulation, and, while Amph increases locomotion in murine models, DAT knockout mice show a paradoxical reduction in locomotor activity after Amph treatment [8]. On the other hand, mice overexpressing DAT exhibit increased locomotor responses to Amph, an increased amount of DA released by Amph, and a preference for Amph at much lower doses (0.2 and 0.5 mg/kg) than wild-type mice (2 mg/kg) [9]. Taken together, these data highlight the primary role played by DAT in mediating Amph-induced effects. However, it remains unclear if chronic exposure to Amph during early development, for example, during embryogenesis, causes functional and/or behavioral consequences in adult animals [10,11]. This is an important issue since abuse of Amph also occurs during pregnancy. While therapeutic doses of Amph do not cause harm in offspring [11], studies in mammal models have suggested that prenatal exposure to psychostimulants like cocaine or methamphetamine can lead to long-lasting alterations in the dopaminergic system, such as changes in DAT density and DA levels, which can persist into adulthood. For example, Leslie et al. demonstrated that prenatal cocaine exposure reduces DAT density in juvenile rats [12], whereas rats exposed to methamphetamine prenatally exhibit higher levels of basal DA and a higher response to challenging doses of methamphetamine [13]. But the mechanisms underlying these long-lasting effects have not yet been elucidated [14]. Moreover, previous publications do not explain if these effects are long-lasting consequences of drug-induced physiological changes in the embryos, which then persist into adulthood, or if drug-induced modifications in maternal care play a role. The latter is an important consideration because several reports show that subtle variations in maternal care reshape offspring development and behaviors via a non-genomic but epigenetic transmission [14,15]. Thus, the behavioral changes reported in juvenile animals exposed to drugs of abuse prenatally could be a consequence of the effects that drugs have on maternal care rather than a direct biologic effect on the embryos which then manifest later in life. To test if Amph causes physiological changes during embryogenesis without the interference of possible effects from Amph-induced changes in maternal care, we used the simple model *Caenorhabditis elegans* (*C. elegans*). This model lacks maternal care but retains a conserved dopaminergic response to Amph [16,17]. In fact, previous data showed that when forced to swim in solutions containing 0.3–1 mM Amph, *C. elegans* exhibits a unique behavior named Swimming-Induced Paralysis (SWIP), and we demonstrated that Amph-induced SWIP is caused by an overload of DA in the synapses through DAT-1 reversal function. The “paralysis” induced by Amph treatments is only temporary because, when Amph is washed out, animals’ ability to swim is restored [16]. As increased levels of extracellular DA can also be achieved by blocking DAT-1, we observed SWIP both in wild-type animals treated with DAT inhibitors and in *dat-1* knockout animals [18]. Moreover, our animal model is also suitable for the investigation of the epigenetic mechanisms underlying the long-lasting effects caused by Amph as histone methylation is one of the most conserved epigenetic modifications between *C. elegans* and mammals [19].

## 2. Results

### 2.1. Chronic Exposure to Amphetamine during Embryogenesis Induces Amphetamine Hypersensitivity in Adult Animals

Previously, we showed that an excess of extracellular DA caused by Amph hindered the ability of *C. elegans* to swim. We named this behavior Swimming-Induced Paralysis, or SWIP [16,20]. Here, we used SWIP to test whether exposure to Amph during early development alters the ability of worms to swim later in life. Embryos collected from synchronized gravid adults were incubated with 0.5 mM Amph dissolved in M9 buffer or M9 buffer alone (control) for 15 h (Figure 1a). This incubation time was chosen because in *C. elegans*, embryogenesis and differentiation of the nervous system are completed in about 14 h, and 0.5 mM Amph was chosen to reproduce the psychostimulant/addictive doses of Amph in mammals [5]. Animals were then washed to carefully remove any residual Amph and grown in food-seeded plates. Two days later, when animals were young adults (late larval-stage L4), we tested their ability to swim in a vehicle solution. No difference was found between animals exposed to Amph during embryogenesis (Embryo—Amph → Adult—Vehicle. Figure 1b, green triangles) with respect to those exposed to control solution (Embryo—Cont→Adult—Vehicle. Figure 1b, black circles). This result shows that chronic Amph exposure during early development does not cause motor dysfunctions in adult animals. Interestingly, when we challenged both groups with 0.5 mM Amph for 10 min, the percentage of animals exhibiting SWIP was significantly higher in those exposed to Amph during embryogenesis (Embryo—Amph→Adult—Amph; Figure 1b, red triangles) with respect to animals pre-exposed to the control solution (Embryo—Cont→Adult—Amph; Figure 1b, blue circles). This result demonstrates that Amph exposure during early development enhances *C. elegans*’ behavioral response to Amph later in life. It is worth pointing out that data shown in Figure 1b were obtained from 72 independent trials. In each trial, 10–50 animals per group were tested. When we graphed each individual trial, we observed no statistical difference between the Embryo—Amph→Adult—Amph and Embryo—Cont → Adult—Amph groups in 14 trials out of 72. This result suggests that following embryonic exposure to Amph, animals have an 80% probability of exhibiting hypersensitivity to Amph when they become young adults. For this reason, all our ex vivo experiments, as shown below, were performed in parallel with SWIP assays.

### 2.2. Chronic Exposure to Amphetamine during Embryogenesis Alters Histone Marks in the dat-1 Gene and Decreases Expression of DAT-1 in Adult Animals

Previous publications have shown that environmental insults, such as alcohol exposure during early development, cause epigenetic changes and alter the expression of specific genes in adult animals [21]. Therefore, we tested if this was the case with Amph. Specifically, we investigated whether Amph exposure during development changed histone marks in genes required for SWIP. We focused our studies on DAT-1, because this protein is a direct target of Amph both in mammals and *C. elegans*, and genetic ablation of DAT prevents Amph-induced behaviors [16,22]. Among the different histone marks, we focused on H3K4me3 (histone 3 trimethylated at lysine 4) and H3K9me2 (histone 3 dimethylated at lysine 9) as previous studies showed that in adult mice or rats, psychostimulants such as methamphetamine or cocaine alter these marks and, consequently, gene expression in the reward system [23,24]. As with our behavioral experiments, animals went through 15 h of exposure to Amph during embryogenesis following the same protocol as in Figure 1a. Two days later, when animals were at the stage of young adults, a small group of animals *(N* = 50 per group) were tested to confirm Amph hypersensitivity (the Embryo—Amph→Adult—Amph group exhibited significantly higher SWIP with respect to the Embryo—Cont→Adult—Amph group; 83 ± 2% and 48 ± 4%, respectively; *p* < 0.003, *t*-test with t = 4 and df = 8). The remaining animals were processed for a chromatin immunoprecipitation (ChIP) assay. To investigate if embryonic exposure to Amph altered the levels of H3K4me3 and H3K9me2 in the *dat-1* gene, ChIP assays using antibodies to these marks were performed. The immunoprecipitated DNA was evaluated using primers upstream of and at the transcription start site (TSS), as well as within the last intron of *dat-1* gene (amplicons indicated as 1, 2 and 3, respectively, in Figure 2a–d). Interestingly, our results show that samples isolated from animals exposed to Amph during embryogenesis exhibited a significant reduction in H3K4me3 both upstream of *dat-1* (from 0.9 ± 0.2 to 0.4 ± 0.02) and at the TSS (from 1.51 ± 0.2 to 0.5 ± 0.2) with respect to samples obtained from control-pretreated animals (Figure 2b; * *p* = 0.001 and ** *p* = 0.0001, respectively; two-way ANOVA). No significant difference between control and treated animals was observed at site 3 of *dat-1* (from 0.6 ± 0.2 to 0.4 ± 0.1) and upstream (from 06 ± 0.2 to 0.6 ± 0.3) of or at the TSS (from 0.5 ± 0.1 to 0.4 ± 0.04) of the house-keeping gene glyceraldehyde phosphate dehydrogenase (*gpdh*; Figure 2b).

H3K4me3 is a transactivation mark, meaning that its enrichment at the promoter of a gene increases the probability that gene is expressed [25]. Similar to H3K4me3, H3K36me3 (histone 3 trimethylated at lysine 36) is also a transactivation mark that is preferentially enriched within the gene body [26]. Thus, we investigated whether H3K36me3 in the *dat-1* gene in young-adult animals was affected by Amph exposure during embryogenesis. As shown in Figure 2c, H3K36me3 was significantly reduced both at site 2 (from 2.7 ± 0.4 to 1.4 ± 0.2) and site 3 (from 2.7 ± 0.4 to 1.4 ± 0.6) of *dat-1* in animals exposed to embryonic Amph with respect to control animals (* *p* = 0.0001; two-way ANOVA), whereas no change was observed between control- and Amph-pretreated samples at the upstream site 1 of *dat-1* (0.6 ± 0.1 and 0.5 ± 0.1, respectively) or at site 1 (1.7 ± 0.2 and 1.7 ± 0.1, respectively) and site 2 (2.4 ± 0.3 and 2.2 ± 0.2, respectively) of the *gpdh* gene. The reduction in H3K4me3 and H3K36me3 suggests that Amph exposure during early development reduces the expression of *dat-1* in adult animals. We speculated that the loss of active histone marks would be accompanied by an increase in histone marks associated with gene silencing. To test this hypothesis, we performed ChIP assays for the repressive mark H3K9me2 known to accumulate around the TSS of repressed genes [27]. Indeed, we found that H3K9me2 was significantly increased from 0.4 ± 0.2 to 0.6 ± 0.2 at the TSS (site 2) of *dat-1* (Figure 2d; * *p* = 0.02, two-way ANOVA). No significant change in H3K9me2 was detected at site 1 (from 0.4 ± 0.2 to 0.6 ± 0.3), downstream at site 3 (from 0.4 ± 0.3 to 0.5 ± 0.3) of *dat-1*, and at site 1 (from 0.5 ± 0.1 to 0.5 ± 0.2) and site 2 (from 0.5 ± 0.1 to 0.4 ± 0.04) of the *gpdh* gene. Taken together, these data demonstrate that Amph exposure during embryonal development changes the epigenetic landscape of the *dat-1* gene in adult animals, possibly leading to a reduction in DAT-1 proteins.

Our ChIP data suggest that *dat-1* is less likely to be expressed in adult animals that receive Amph during embryogenesis. To verify this observation, we quantified DAT-1 proteins in adult animals. Animals went through the same embryonic treatment as shown in Figure 1a, and when they were at the stage of young adults, worm lysates were collected to perform immunoblots. As shown in Figure 3, animals exposed to Amph during embryogenesis showed significantly reduced levels of DAT-1 with respect to control-exposed animals (* *p* = 0.02, *t*-test with t = 6.84 and df = 4). No change in actin was measured in the same samples stained with the actin antibody (inset in Figure 3). Notably, these results are in agreement with previous studies showing that rats with prenatal exposure to cocaine exhibited a significant decrease in DAT as young adults [12].

### 2.3. The Long-Term Effects Caused by Amphetamine during Embryogenesis in Adult Animals Are Blocked by Genetic or Pharmacological Inhibition of Specific Lysine 9 Methyltransferases

Previous data showed that H3K9-methylated chromatin plays an important role in cell differentiation and neurogenesis during early development [28]. Because our ChIP data show a significant increase in H3K9me2 in the *dat-1* gene in adult animals exposed to Amph during early development (Figure 2d), we investigated whether genetic or pharmacological inhibition of H3K9me2 production prevents the long-lasting behaviors caused by embryonal exposure to Amph. In mammals, five H3K9-specific methyltransferases exist: SUV39H1 and 2, SETDB1, G9A and GLP [29]. These enzymes sequentially add methyl groups to the ε-nitrogen of lysine in position 9 of histone 3 (H3K9). Specifically, G9A mediates mono- and dimethylation of H3K9 [30], whereas SETDB1 mediates di- and trimethylation of H3K9 [31]. Likewise, in *C. elegans*, the SETDB1 homolog MET-2, which is involved in heterochromatin formation [32], deposits mono- and dimethyl groups on H3K9, while SET-25, a G9A homolog required for the establishment of long-term silencing signals, is responsible for H3K9 di- and trimethylation [33]. We hypothesized that genetic or pharmacological ablation of SET-25 or MET-2, i.e., changes in H3K9me2 availability, would prevent the long-term effects caused by Amph. We found that adult *set-25* knockout animals did not exhibit Amph hypersensitivity caused by embryonal exposure to Amph (compare the red triangles in Figure 4a with Figure 1b). On the contrary, a significant reduction in Amph-induced SWIP was observed in adult *set-25* mutants challenged with Amph after embryonic exposure to Amph with respect to animals exposed to control solution during embryogenesis (red triangles and blue circles, respectively, in Figure 4a. *p* = 0.002, q = 5.4 two-way ANOVA Tukey multiple comparison test). And, as in wild-type animals (Figure 1b), no SWIP was observed in adult *set-25* mutants challenged with vehicle and pretreated with control or Amph (open circles and green triangles, respectively, in Figure 4a).

Next, we tested if pharmacological inhibition of H3K9 methyltransferase prevented the long-lasting effects caused by embryonal Amph exposure by using the G9A/MET-2 specific inhibitor, UNC0224. Embryonic treatments with 1 µM UNC0224 alone did not compromise the ability of wild-type animals to swim in the vehicle (gray squares in Figure 4b), and as shown in Figure 1b, embryonal exposure to Amph caused Amph hypersensitivity in adult animals with respect to animals pretreated with control solution (red triangles and blue circles, respectively, in Figure 4b; *p* < 0.0001 two-way ANOVA). However, adult animals challenged with Amph for 10 min after co-treatment with 1 µM UNC0224 and 0.5 mM Amph during embryogenesis (brown squares in Figure 4b) exhibited significantly lower rates of SWIP than adult animals challenged with Amph after embryonal exposure to Amph alone (red circles in Figure 4b; *p* < 0.0001, two-way ANOVA). Because the Embryo—UNC0224 + Amph→Adult—Amph animals (brown squares) exhibited a curve for SWIP comparable to that of animals exposed to control solution during embryogenesis (blue circles in Figure 4b), these data indicate that UNC0224 blocks Amph hypersensitivity measured after embryonic Amph exposure. Taken together, these results show that the histone methyltransferases SET-25 or MET-2 are required by Amph to induce long-lasting behavioral effects. Thus, methylation at lysine 9 of histone 3 is an essential step to consolidate the long-lasting behavioral effects initiated by chronic embryonal exposure to Amph.

## 3. Discussion

To identify the behavioral and functional effects caused by embryonic exposure to Amph in adult animals, we used the simple model *C. elegans*. This model allowed us to measure the physiologic consequences of embryonal exposure to psychostimulants without the interference of potential effects of the drugs on maternal care. Because *C. elegans* embryos can be isolated at early stages and can develop independently, our embryonic treatments occurred with no functional and/or behavioral contribution from parental lines. This ensured that the effects observed in adult animals were the result of long-term modifications induced by Amph in the embryos rather than acquired modifications originating from the effects Amph had on the parental lines. As recent human data showed that therapeutic concentrations of Amph taken during pregnancy do not affect neurodevelopment and growth in offspring [11], we investigated only the effects of high doses of Amph (0.5 mM) which were previously shown to cause reliable behavioral and functional effects in *C. elegans* [16,34] and mammals [22].

Our data demonstrate that chronic Amph exposure during development generated adult animals that are hypersensitive to Amph. In fact, saturating levels of Amph-induced behaviors occurred within shorter treatments (~6 min; Figure 1b) in animals exposed to Amph during embryogenesis with respect to animals exposed to control solution (>10 min; Figure 1b). Moreover, after 10 min of Amph exposure, the number of young adults exhibiting SWIP was significantly higher in the group of animals which received Amph during embryogenesis (Figure 1b). These results strongly suggest that Amph exposure during early development induces physiological changes that are maintained after embryogenesis and manifest later in life. Interestingly though, these changes were evident only when adult animals were challenged with Amph. No obvious functional or behavioral modification was observed in animals crawling on plates or swimming in the vehicle. This suggests that the long-lasting effects caused by embryonic Amph exposure occur in genes recruited by Amph e.g., *dat-1*.

SWIP is a well-characterized behavior of *C. elegans*. It occurs with excessive synaptic levels of DA. As matter of fact, genetic and pharmacological ablation of *dat-1* causes SWIP in the absence of Amph [18,20], whereas animals unable to synthetize DA exhibit no Amph-induced SWIP [16]. For these reasons, we investigated whether DAT-1 plays a role in the long-lasting behavioral effects seen following chronic exposure to Amph during early development. Our data show a significant reduction in the protein expression of DAT-1 in adult animals exposed to Amph during embryogenesis (Figure 3). In these animals, we can assume that reduced levels of DAT-1 caused an increase in DA in the synaptic cleft and, while in normal conditions this increase in DA is not enough to generate a phenotype (Figure 1b, green triangles), when animals were challenged with Amph, we measured an increase in SWIP (Figure 1b, red triangles). Thus, our data suggest that embryonic Amph exposure reduces the number of DATs and ultimately causes Amph hypersensitivity (Figure 1b).

A growing body of evidence has demonstrated that drugs of abuse influence gene expression. These data support the hypothesis that drug-induced epigenetic adaptations are among the main processes through which drugs induce long-term changes, which ultimately result in addiction [35]. For these reasons, we investigated whether the behavioral and functional effects seen in adults exposed to Amph during embryogenesis originated from stable epigenetic alterations in genes required by Amph to induce behavioral effects. We investigated changes in the gene expression of the DA transporter, DAT-1, because besides being a direct target of Amph both in mammals and *C. elegans* [16,22], DAT plays a critical role in maintaining basal levels of DA in the synaptic cleft. Among the different epigenetic mechanisms, we chose to study histone methylation because it is highly conserved between humans and *C. elegans* [29] and because histone methylation plays a major role in regulating gene expression during early development [30]. We focused on histone methylation at lysine (K) 4, 9 and 36 (H3K4me3, H3K9me3 and H3K36me2) as previous studies showed that drugs of abuse, such as cocaine and methamphetamine, modify the landscape of these marks in specific genes of the reward system [23,24]. In agreement with our expression data (Figure 3), our ChIP data show an increase in the repression mark H3K9me2 and a decrease in the activation marks H3K4me3 and H3K36me3 in the *dat-1* gene following chronic exposure to Amph during embryogenesis (Figure 2). These results suggest that histone methylation is a mechanism involved in the long-lasting effects generated by embryonal exposure to Amph.

Recent data suggest that methamphetamine use during pregnancy impairs differentiation of the ventral midbrain dopaminergic neurons by altering the expression of relevant genes involved in neurogenesis [36]. Moreover, recent studies demonstrated that H3K9me2-mediated silencing is an important epigenetic regulator of gene expression throughout development [28], while a reduction in H3K9me2 levels and downregulation of G9A, a specific lysine 9 methyltransferase (K9MT), has been observed in the brain of mice repeatedly exposed to cocaine [37]. For these reasons, we further investigated the role of H3K9me2 in the long-lasting effects of embryonal exposure to Amph by depleting the production of H3K9me2 both pharmacologically and genetically. Five enzymes with specific K9MT activity have been identified in mammals, while in *C. elegans* only two K9MTs exist: SET-25, which is the homologue of human G9A and SUV39H, and MET-2, which is the homologue of the human SETDB1 [28]. Our results show that genetic ablation of *set-25* prevents the Amph hypersensitivity seen in adults exposed to Amph during embryogenesis (Figure 4a). Moreover, co-exposure to Amph and the specific G9A inhibitor, UNC0224, during embryogenesis blocked Amph hypersensitivity in wild-type adults (Figure 4b). These data strongly suggest that H3K9me2 is needed for the long-lasting behavioral modifications caused by embryonal exposure to Amph.

In conclusion, our data demonstrate that chronic exposure to Amph during embryogenesis causes stable epigenetic and physiological changes in DAT-1 and the effects of these modifications manifest later in life when animals are challenged with Amph. These results are in agreement with mammalian data reporting that adult mice with prenatal cocaine exposure exhibit a reduction in DAT in the striatum [12]. It is possible, therefore, that different psychostimulants induce similar long-lasting modifications. The murine data [12] and the results reported in this manuscript support the idea that the dopaminergic response to psychostimulants is highly conserved between *C. elegans* and mammals [38]. Notably, our study identifies, for the first time, histone methylation as a possible mechanism employed by Amph to perpetuate functional and behavioral modifications after chronic embryonal exposure. As epigenetic mechanisms are often highly conserved between organisms [29], studying the epigenetic modifications induced by Amph in *C. elegans* has the potential to inform our investigations in mammals.

## 4. Materials and Methods

### 4.1. Worm Husbandry and Strains

Animals were grown at 20 °C in NGM plates seeded with N22 bacteria in non-crowded conditions. The N2 wild-type (Bristol variety), *met-2(n4256)III* and *set-25(n5021) III* were purchased from the Caenorhabditis Genetic Center (CGC), University of Minnesota.

### 4.2. Embryonic Exposure to Amphetamine

Embryos were released from gravid worms by treating the worms for 5–10 min with a solution containing 10 N NaOH and 10% sodium hypochlorite. After 3 washes with *C. elegans* egg buffer and spinning for 3 min at 1200 rpm, embryos were separated from charkas in a 30% sucrose gradient and centrifuged at 1200 rpm for 5 min. Floating embryos from the solution meniscus were collected and washed 3 times with sterile water at 1200 rpm. Isolated embryos were then incubated for 15 h with 0.5 mM Amph alone, 0.5 mM Amph and 1 µM UNC0224 (Cayman Chemical—BA E-6300R), 1 µM UNC0224 alone or control solution (M9 buffer). Larvae at stage 1 (L1) were washed 3 times in M9 buffer and seeded in NGM plates containing NA22 *E. Coli* bacteria and grown at 20 °C. Approximately 48 h later, when animals achieved the stage of late L4 (young adults), they were tested for SWIP or processed for chromatin or protein extraction for ChIP and Western blot experiments, respectively.

### 4.3. Swimming-Induced Paralysis (SWIP) Assays

In each SWIP trial, 8–16 animals in late larva stage 4 (young adults) were transferred into 40 μL of vehicle (200 mM sucrose, which guarantees physiological osmolarity) with or without 0.5 mM Amph (NIDA, Research Triangle Institute) in a single well of a Pyrex spot plate (Thermo Fisher Scientific, Waltham, MA, USA). Paralyzed animals were counted every minute for 10 min using an inverted microscope (Carl Zeiss, Inc., Thornwood, NY, USA). The number of paralyzed animals was reported as a percentage of the total number of animals observed in each trial ± standard error.

### 4.4. Chromatin Immunoprecipitation (ChIP) Assays

After 3 washes with M9, late-L4 animals were collected in ice-cold PBS with proteinase and phosphatase inhibitors (Thermo Fisher). Using a 1 mL pipette, animals were dropped into a 50 mL glass beaker containing liquid nitrogen to flash-freeze worms into ice balls. Frozen worms were disintegrated into powder using a hammer homogenizer. The worm powder was fixed with 1.1% formaldehyde in PBS with proteinase and phosphatase inhibitors (ThermoFisher) at room temperature for 10 min. The reaction was quenched by adding 125 mM glycine for 5 min. Samples were washed one time with PBS at 4000× *g* for 3 min and two times with Dounce buffer (Sucrose 0.35 M, HEPES-KOH pH 7.5 15 mM, EGTA 0.5 mM, MgCl_2_ 0.5 mM, KCl 10 mM, EDTA 0.1 mM, DTT 1 mM, Triton X-100 0.5%, NP-40 0.25%) at 16,000× *g* for 5 min. Chromatin collected from the top aqueous clear meniscus was fragmented with micrococcal nuclease (10,000 units/mL) to mostly mononucleosomal fragments and incubated with antibodies against H3K4me3, H3K36me3 or H3K9me2 (0.025 μg/μL) overnight. The samples were then incubated with 40 μL of pre-washed protein A beads plus salmon sperm DNA (4 ng/mL) at 4 °C for 3 h. The immunoprecipitated chromatin was incubated at 65 °C overnight to reverse crosslinking, treated with proteinase K (100 μg/mL) and RNase A (200 μg/mL) and subsequently used as template for qPCR with primers for amplification of specific regions after purification. Three sets of primers, labeled 1, 2 and 3, were designed for *dat-1*. Primer set 1 amplified the region from position −900 to −788 and primer sets 2 and 3 amplified the region between positions −81 to +125 and +3048 to +3218, respectively. Two sets of primers were designed for the house-keeping gene *gpdh*. Primer set 1 amplified the region between position −27 to +149 and primer set 2 amplified the site in position +1436 to +1615. The reaction procedure was as follows: 95 °C for 3 min, followed by 41 cycles at 95 °C for 10 s and 60 °C for 30 s, and lastly, 98 °C for 10 s and 65 °C for 31 s, followed by a melt curve from 65 °C to 95 °C. Data were analyzed using the 2^−ΔΔCt^ method. For this method, the average Ct value of act-1, the *C. elegans* actin homolog, was calculated for each sample and used as a control. Then, each Ct value for the genes of interest was subtracted by the actin Ct value average. The product of that subtraction was then subtracted by the average of the “calibrator” (control sample). Finally, the value of the target–calibrator was expressed as 2^−x^, where x represents the target–calibrator value. The final value represents fold change with respect to the “calibrator”.

### 4.5. Western Blot

Protein lysates were extracted from about 5000 worms with RIPA buffer supplemented with proteinase and phosphatase inhibitors followed by three freeze/thaw cycles in liquid nitrogen. The concentration of proteins was quantified with a BCA protein assay kit (Thermo). The protein samples were separated in SDS-PAGE and transferred to PVDF membrane (Immobilon^®^-P, Millipore, Burlington, MA, USA) for immunoblotting. An amount of 60 μg of protein was loaded and blotted with primary antibody against DAT-1 antibody (a gift from Dr. Randy Blakely). Actin antibody (MAB1501, Millipore) was used as the loading control. The image was developed with the second antibody conjugated with IRDye (IRDye 800CW, IRDye 680RD, LI-COR). The intensity of protein bands was quantified and analyzed with ImageJ 1.5.3 software.

### 4.6. Statistical Analysis

GraphPad Prism software 7 (GraphPad Software, Inc., San Diego, CA, USA) was used for statistical analyses. The statistical significance was determined using one-way or two-way ANOVA with multiple comparison tests and Student’s t-tests. The SWIP data passed the Shapiro–Wilk normality test (α = 0.05). Data are reported as averages of at least 3 independent experiments ± SEM or SD.

## Figures and Tables

**Figure 1 ijms-24-13092-f001:**
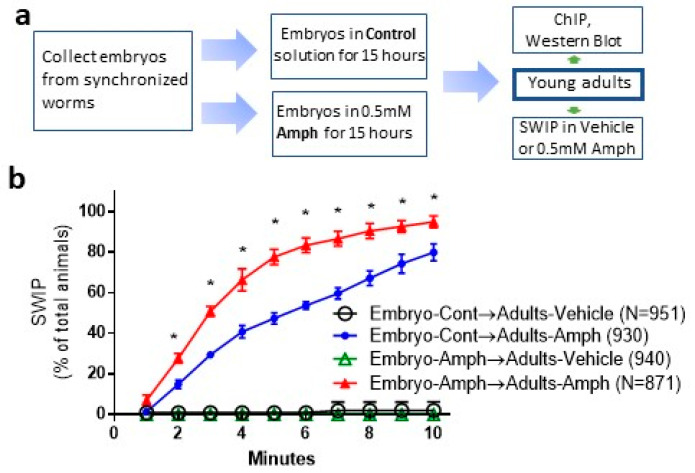
Embryonic exposure to Amph increases Amph-induced behaviors in adult animals. (**a**) experimental design to examine functional, physiological and behavioral consequences of embryonic Amph treatments. (**b**) Animals pretreated with 0.5 mM Amph during embryogenesis (red triangles) exhibit higher Amph-induced SWIP with respect to animals pretreated with control solution (blue circles) * *p* < 0.0001, two-way ANOVA Tukey’s multiple comparisons with Interaction Factor: F(27, 360) = 47.36 and Treatment Factor: F(3, 360) = 2062. Data are average of 72 independent trials and are expressed as percentage of animals exhibiting paralysis ± SEM.

**Figure 2 ijms-24-13092-f002:**
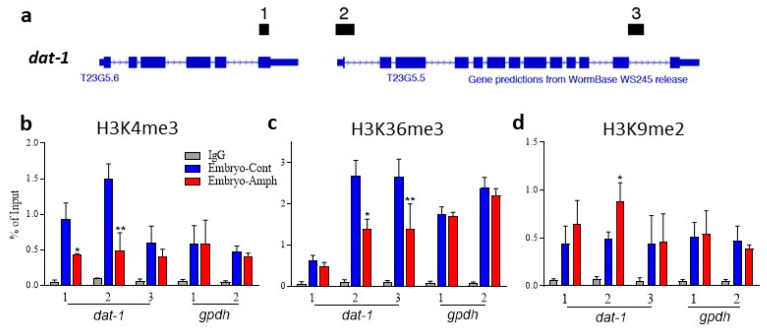
Embryonic Amph exposure changes histone marks in the *dat-1* gene in adult animals. (**a**) graphic illustration of 3 primer sets (1, 2, 3) used to amplify gene regions of *dat-1* for ChIP experiments. (**b**) Statistical significance was measured for H3K4me3 in *dat-1* amplicon 1 (* *p* = 0.001; q = 5.4) and 2 (** *p* < 0.0001; q = 10.7) after embryonic treatment with Amph (red bars) with respect to controls (blue bars). Interaction Factor: F(8, 30) = 6.35 and Treatment Factor: F(2, 30) = 81.6. No significant change was measured in *dat-1* amplicon 3 (*p* = 0.3; q = 2) and the *gpdh* amplicons 1 and 2 (*p* > 0.9; q = 0.02 and *p* = 0.8; q = 0.6, respectively). (**c**) Statistical significance was measured for H3K36me3 at *dat-1* amplicon 2 (* *p* < 0.0001; q = 8.9) and 3 (** *p* < 0.0001; q = 8.7) after embryonic treatment with Apmh. Interaction Factor: F(8, 30) = 11.9 and Treatment Factor: F(2, 30) = 232. No significant change was measured at the *dat-1* amplicon 1 (*p* = 0.7; q = 1) and the *gpdh* amplicon 1 and 2 (*p* = 0.9; q = 0.2 and *p* = 0.6; *q* = 1.2, respectively). (**d**) Statistical significance was measured for H3K9me2 in the *dat-1* amplicon 2 (* *p* = 0.02; q = 4) after embryonic treatment with Amph. Interaction Factor: F(8, 30) = 1.3 and Treatment Factor: F(2, 30) = 41.8. No significant change was measured in *dat-1* amplicon 1 (*p* = 0.3; q = 2.1) and 3 (*p* = 0.9; q = 0.2) and the gpdh amplicons 1 and 2 (*p* = 0.9; q = 0.3 and *p* = 0.8; q = 0.8, respectively). Each graph shows the average of 3 independent experiments and data are presented as ± SD. Statistical analysis was performed using two-way ANOVA Tukey’s multiple comparisons test.

**Figure 3 ijms-24-13092-f003:**
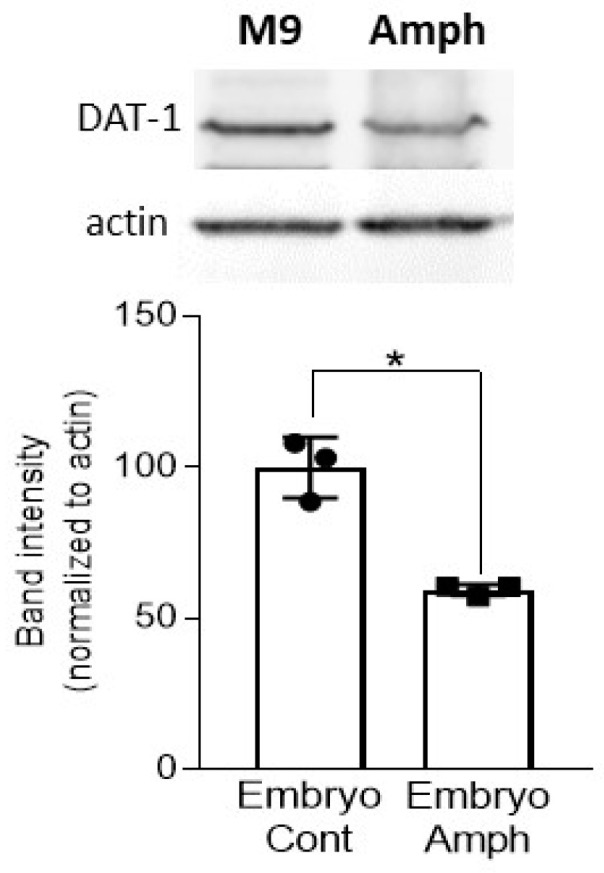
Embryonic Amph exposure changes DAT-1 expression in adult animals. Western blot experiments show reduced DAT-1 proteins in adults that received Amph during embryogenesis (* *p* = 0.002; t = 6.84, df = 4), whereas no change in actin was observed in the same samples. Inset shows images of one representative blot image from three independent experiments. Statistical analysis was performed using Student unpaired two-tailed *t*-test.

**Figure 4 ijms-24-13092-f004:**
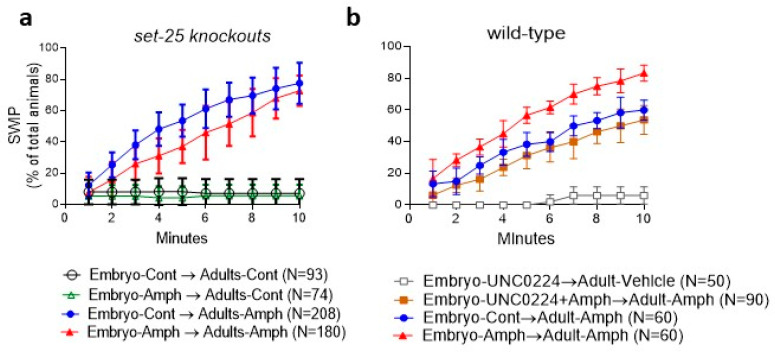
MET-2/G9A and SET-25/SETDB1 are required during embryonal exposure to Amph to induce long-lasting behavioral effects in adult animals. (**a**) *set-25* knockout animals pretreated with 0.5 mM Amph during embryogenesis (red triangles) exhibit a significant reduction in Amph-induced SWIP with respect to control-pretreated animals (blue circle) (*p* = 0.002; q = 5.4. Interaction Factor: F(27, 414) = 34.7 and Treatment Factor: F(3, 46) = 67.5). (**b**) wild-type adult animals pretreated with 0.5 mM Amph and 1 µM UNC0224, a MET-2/G9A inhibitor (brown square), exhibit a significant decrease of Amph-induced SWIP with respect to animals pretreated with Amph alone (red circles; *p* < 0.0001, q = 27.9 with Interaction Factor F(27, 210) = 8.2 and Treatment Factor F(3, 210) = 530). Adult animals pretreated with Amph alone during embryogenesis exhibit significant increase of Amph-induced SWIP with respect to animals pretreated with control solution (compare red triangles with blue circles; *p* < 0.0001, q = 18.3). Pretreatments with UNC0224 alone during embryogenesis do not affect the ability of adult worms to swim in the vehicle (grey squares). Data in **a** and **b**, which are average of 3 and 4 independent experiments respectively, are expressed as percentage of animals exhibiting paralysis ± SEM. Statistical analysis was performed using two-way ANOVA Tukey’s multiple comparison test.

## Data Availability

All data generated in this manuscript are included in this manuscript.

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
