# Peer review of "Long-Lasting Epigenetic Changes in the Dopamine Transporter in Adult Animals Exposed to Amphetamine during Embryogenesis: Investigating Behavioral Effects"

_ijms, 2023, doi:10.3390/ijms241713092_

Round 1

Reviewer 1 Report

1.     Introduction does not provide much rationale for the study. For example, lines 60-73 is all summary of the results. Please replace with justification of the experimental approach, especially on the dose of amphetamine used - the only rationale given was 'addictive dose of amphetamine'.

2.     F or t values, degrees of freedom for significant and non-significant findings are missing (i.e., Figures 1,2, and 4 ANOVAs, each factor and interaction should be clearly stated, with Bonferroni corrections for post hoc multiple comparisons clearly stated). It is unclear what all the p values are referring to. For t-tests do report the t values.

3.     Figure 2 should show individual values, significance in 2d is particularly unconvincing.

4.     Figure 3 – compared to the original images file, the bands width for actin has been dramatically reduced.

5.     Figure 3 is hugely underpowered, which can lead to type 1 error.

6.     Consistent with the comment #4, Figure 4 shows that the lack of embryonic amphetamine exposure effects due to met-2 or set-25 knockouts are driven by the control exposure groups moving significantly more than wildtype, compared to Figure 1 and 4C.

7.     Discussion is not supported by the results.  

Author Response

We thank reviewer-1 for the meticulous work in reviewing our manuscript. We highly appreciate the suggestions and corrections which improved the quality of the manuscript. Below is point by point response to the Reviewer:

1 – We have removed lines 60-73 and added text in the introduction to provide more rationale for the study and justification about the dose of Amph used (see text in red in the revised version of the manuscript)

2 – F or t values for significant and non-significant findings have been included int the Result section and Legends for Figures 1, 2, 3 and 4.

3 - Individual values for Figure 2 have been included in the Result section 2.2 and in the Legend of Figure 2. Statistical significance in Figure 2d is measured with 2way-ANOVA Tukey’s multiple comparisons test (see Figure Legend 2 for statistics details).

4 – The Western Blot image in Figure 3 is cropped from the original gel blotted against DAT-1 antibody. No actine band was shown in the Western blot image originally submitted. The original Western blot bands for actin are included below:

5 - Figure 3 is the average of 3 independent experiments where each experiment was performed with three replicants of control-treated samples and three replicants of Amph-treated samples. For this reason, we are confident that data presented in this figure are not underpowered.

6 – We agree with the reviewer that Figure 4a, relative to met-2 mutants, deserves further investigation. As at this point, we can’t explain why met-2 mutants exhibit higher Amph-induced SWIP than wildtype animals, we removed this data from the manuscript.

7 – We have restricted our discussion only to the data presented in this manuscript.

Reviewer 2 Report

Thank you for the opportunity to review the article by Ke et al. titled: "Long-lasting Epigenetic Changes at the Dopamine Transporter Cause Behavioral Effects in Adult Animals Exposed to Amphetamine During Embryogenesis". 

This article shows that chronic embryonic exposures to amphetamine can alter the expression of DAT-1 gene in C. elegans nematodes. Despite being a very complete article. I include some comments to take in consideration:

Please, DAT-1 protein must be in capital letters all the time, and when talking about the gene, it must be written in italics.

The introduction does not include the objectives of the project.

Line 59-73: are results of the study, they should not go in the introduction.

Despite being a new topic, update the references with results of the same recent nature.

Line 302-307: The final conclusion is very ambitious for the results obtained. More studies with samples in mammals and humans and replicates are required to reach this conclusion.

Author Response

We thank Reviewer-2 for the meticulous work in reviewing our manuscript. We highly appreciate the suggestions and corrections which improved the quality of the manuscript. Below is point by point response to the Reviewer:

1 – DAT-1 and dat-1 have been corrected to reflect protein and gene, respectively.

2- Introduction has been modified to include the objective of the project (see red text in the revised manuscript)

3- Lines 59-73 have been removed from the Introduction.

4- New references have been included (see lines 270-272)

5- Lines 302-307 have been removed from the final conclusions

Reviewer 3 Report

Ke and colleagues in the present research article entitled ‘Long-lasting Epigenetic Changes at the Dopamine Transporter Cause Behavioral Effects in Adult Animals Exposed to Amphetamine During Embryogenesis’ investigated the long-lasting effects of amphetamine exposure during embryogenesis on behavioral and physiological changes in adult animals using the model organism Caenorhabditis elegans (C. elegans). To understand if amphetamine exposure effects on dopamine transporter density also occur in the absence of maternal care, the researchers used C. elegans, which lacks maternal care but retains a conserved dopaminergic response to amphetamine. They exposed C. elegans embryos to amphetamine and observed that the young adults showed hypersensitivity to amphetamine. This suggests that the exposure during early development induces long-lasting changes that manifest later in life. Further investigation revealed that embryonic amphetamine exposure led to epigenetic changes at the DAT gene (dat-1) in young-adult worms. Specifically, the researchers observed alterations in histone marks associated with active or repressed transcription (H3K4me3 or H3K9me2, respectively) and transcription elongation (H3K36me3) at the dat-1 gene. These changes correlated with decreased expression of DAT-1 protein in adult animals. To understand the role of histone methylation in these long-lasting effects, the authors inhibited specific lysine 9 methyltransferase enzymes, which are involved in depositing H3K9me2 marks. Genetic or pharmacological inhibition of these enzymes prevented the amphetamine hypersensitivity in adult animals, suggesting that H3K9me2 is crucial for consolidating the long-lasting effects of embryonic amphetamine exposure. Overall, this study demonstrates that amphetamine exposure during embryogenesis induces stable epigenetic and physiological changes in C. elegans, resulting in amphetamine hypersensitivity in adult animals. The findings highlight the potential role of histone methylation in the long-lasting effects of drug exposure during early development.

In general, I think the idea of this article is really interesting and the authors’ fascinating observations on this timely topic may be of interest to the readers of International Journal of Molecular Sciences. However, some comments, as well as some crucial evidence that should be included to support the author’s argumentation, needed to be addressed to improve the quality of the manuscript, its adequacy, and its readability prior to the publication in the present form.

Please consider the following comments:

• I recommend revising the title. In its current form, I find it to be quite long and contains specific scientific terms that may be challenging for the average reader to understand at first glance. Furthermore, the title mentions "Adult Animals Exposed to Amphetamine During Embryogenesis." It's not immediately clear whether the focus is on the effects of amphetamine exposure during embryogenesis or the long-lasting epigenetic changes. Suggestions: "Long-term Epigenetic Changes at the Dopamine Transporter in Adult Animals Exposed to Amphetamine During Embryogenesis: Investigating Behavioral Effects" [1-3].

• A graphical abstract that will visually summarize the main findings of the manuscript is highly recommended.

• Abstract: According to the Journal’s guidelines, this section should be presented as a short summary of about 200 words maximum that objectively represents the article. It should let readers get the gist or essence of the manuscript quickly, prepare the readers to follow the detailed information, analyses, and arguments in the full paper and, most of all, it should help readers remember key points from your paper. Please, consider rewriting this paragraph following these instructions [4]. 

• Keywords: Please list ten keywords chosen from Medical Subject Headings (MeSH) and use as many as possible in the title and in the first two sentences of the abstract. I would suggest adding “Epigenetic modifications”, “C. elegans” and “Neural development” as keywords.

• Introduction: The authors need to reorganize this section with several paragraphs made up of about 1000 words, introducing information on the main constructs of this study, which should be understood by a reader in any discipline, and making it persuasive enough to put forward the main purpose of the current research the author has conducted and the specific purpose the author has intended by this protocol. I would like to encourage the authors to present the introduction starting with the general background, proceeding to the specific background on the drug amphetamine (Amph) and its effects on the central nervous system, and finally the current issue addressed to this study, leading to the objectives. Those main structures should be organized in a logical and cohesive manner [5]. 

• In this regard, I believe that the Introduction section would benefit from additional information to enhance its clarity and contextualization. To strengthen this section, I suggest highlighting the understanding of the neural bases underlying the effects of amphetamine exposure during embryogenesis by including relevant information about the dopaminergic system and its significance in mediating the behavioral and physiological responses to amphetamine. The neural bases underlying the effects of amphetamine exposure during embryogenesis can be attributed to the drug's interaction with the dopaminergic system, which plays a crucial role in regulating reward, motivation, and motor functions. Amphetamine exerts its psychostimulant and addictive properties primarily by increasing the extracellular levels of dopamine (DA) in the brain. The dopamine transporter (DAT) is a key protein responsible for the reuptake of synaptic dopamine, maintaining its homeostasis and terminating its signaling. Previous studies have extensively investigated the role of DAT in mediating amphetamine-induced behaviors, highlighting its significance in the rewarding and locomotor effects of the drug. Additionally, research in mammalian models has demonstrated that prenatal exposure to psychostimulants like cocaine and methamphetamine can lead to long-lasting alterations in the dopaminergic system, such as changes in DAT density and dopamine levels, which can persist into adulthood [6-7]. Given the conserved dopaminergic response to amphetamine in both mammals and the simple model organism, Caenorhabditis elegans, understanding the epigenetic modifications and changes in gene expression at the DAT gene (dat-1) following embryonic amphetamine exposure can provide valuable insights into the neural mechanisms underlying the long-lasting effects of the drug [8-10]. Understanding the neural bases of amphetamine exposure during embryogenesis could have significant implications for understanding the long-term consequences of early drug exposure and may contribute to developing targeted interventions for drug-related disorders and addiction in human populations.

• Results: In the subsection "2.1. Chronic exposure to amphetamine during embryogenesis induces amphetamine hypersensitivity in adult animals," it would be helpful to provide more context and background information on the SWIP assay and its relevance to amphetamine-induced behavioral effects. Furthermore, when presenting ChIP data in Figures 2b-d, it would be helpful to include error bars or a statistical measure of variability to assess the significance of the changes in histone marks at the dat-1 gene.

• Discussion: The discussion section lacks a clear and structured organization. I would like the authors to begin this section with an introduction and then provide a summary of the previous section. Then, I expect the authors to develop arguments clarifying the potential of this study as an extension of the previous work, the implication of the findings, how this study could facilitate future research, the ultimate goal, the challenge, the knowledge and technology necessary to achieve this goal, the statement about this field in general, and finally the importance of this line of research. It is particularly important to present its limits, its merits, and the potential translation of this protocol into clinical practice. Particularly, the study's relevance to human health and the potential translational implications should be more explicitly discussed. How can the findings in C. elegans inform our understanding of amphetamine exposure during human embryogenesis and its effects on behavior and health?

• Material and Methods: I believe that this section would benefit from a clearer structure and better organization of the flow of information. For example, I would suggest to clearly describe the rationale for choosing Sprague-Dawley rats as the animal model for the study, as well as mention the randomization method used to assign animals to different groups (repeated hypoxia vs. repeated normoxia). Also, here authors should provide a justification for the sample sized used, and include a power analysis or a rationale determining the appropriate sample size to ensure that the study is adequately powered to detect meaningful effects. Additionally, there is no mention of blinding procedures, which are important for minimizing bias during data collection and analysis.

• In my opinion, the ‘Conclusions’ paragraph would benefit from some thoughtful as well as in-depth considerations by the authors, because as it stands, it lists down all the main findings of the research, without really stressing the theoretical significance of the study. Authors should make an effort, trying to explain the theoretical implication as well as the translational application of their research.

• In according to the previous comment, I would ask the authors to include a proper and defined ‘Limitations and future directions’ section before the end of the manuscript, in which authors can describe in detail and report all the technical issues brought to the surface.

• Finally, I would suggest to better address any ethical considerations related to the use of animal subjects in the study and describe the measures taken to ensure the welfare and ethical treatment of the piglets throughout the experimental procedures.

I hope that, after these careful revisions, the manuscript can meet the Journal’s high standards for publication. I am available for a new round of revision of this article. 

Best regards,

Reviewer

References: 

1. https://plos.org/resource/how-to-write-a-great-title/

2. https://www.nature.com/nature-index/news-blog/how-to-write-a-good-research-science-academic-paper-title

3. https://www.indeed.com/career-advice/career-development/catchy-title

4. https://www.mdpi.com/journal/ijms/instructions

5. https://dept.writing.wisc.edu/wac/writing-an-introduction-for-a-scientific-paper/

6. DOI: 10.17219/acem/165944 

7. https://doi.org/10.3390/ijms24065926

8. DOI: 10.3390/biomedicines11030945

9. https://doi.org/10.3389/fnmol.2023.1217090

10. https://doi.org/10.3390/biomedicines11051248

Minor editing of English language required.

Author Response

We thank the Reviewer for the meticulous work in reviewing our manuscript. We highly appreciate the suggestions and corrections which improved the quality of the manuscript. We have addressed each concern/suggestion point by point:

1 – Title has been revised as suggested by the reviewer.

2 – Graphical abstract has been included

3 – Abstract has been revised

4 – keywords have been added

5 – Introduction has been modified as suggested

6 – SWIP assay has been further explained (see lines 74-32). Legend of Figure 2 has been modified to include detailed statistical information. Error bars are more visible in Figure 2b-d.

 7 – Discussion has been modified as suggested

8 – We did not use Sprague-Dawley rats in our study

9 - the translational application, though presented in the original manuscript as “speculation” of our data, has been removed as suggested by reviewer 2. A short sentence has been inserted instead (see lines 27-0-272).

Our research was performed using C. elegans. No ethical or animals’ justifications are required.

Round 2

Reviewer 1 Report

Thank you for the reminder. I have rejected the paper in the first review and would like to not review the paper again.

Thank you for the reminder. I have rejected the paper in the first review and would like to not review the paper again.

Author Response

We do not believe the concerns raised by the Reviewer justify the rejection of our manuscript. 

Reviewer 3 Report

The authors did an excellent job clarifying all the questions I have raised in my previous round of review. Currently, this paper entitled ‘Long-term Epigenetic Changes at the Dopamine Transporter in Adult Animals Exposed to Amphetamine During Embryogenesis: Investigating Behavioral Effects’ is a well-written, timely piece of research that described how amphetamine exposure during embryogenesis induces stable epigenetic and physiological changes in C. elegans, resulting in amphetamine hypersensitivity in adult animals. 

Overall, this is a timely and needed work. It is well researched and nicely written, therefore I believe that this paper does not need a further revision, therefore the manuscript meets the Journal’s high standards for publication.

I am always available for other reviews of such interesting and important articles.

Thank You for your work, Reviewer

Author Response

we thank again the Reviewer for her/his valuable suggestions.